# Harambee! 2.0: Community resources and resilience factors to leverage for improving HIV testing behaviors among African immigrant communities in Seattle, Washington

**Shukri Ahmed Hassan**[1,¤,‡,*], **Najma Sheikh**[2‡], **Guiomar Basualdo**[3], **Nahom Daniel**[4], **Ahmed Ali**[4,5], **Rahel Schwartz**[6,7], **Beyene Tewelde Gebreselassie**[8], **Farah Mohamed**[4,5], **Mohamed Shidane**[5], **Sophia Benalfew**[6], **Bethel Tadesse**[6], **Hirut Amsalu Libneh**[6], **Kifleyesus Bayru**[8], **Yikealo K. Beyene**[8], **Luwam Gabreselassie**[8], **Deepa Rao**[2], **Roxanne P. Kerani**[4‡], **Rena C. Patel**[1‡]

**1** University of Alabama, Birmingham, Alabama, United States of America, **2** Department of Global Health, University of Washington, Seattle, Washington, United States of America, **3** Department of Anthropology, University of Washington, Seattle, Washington, United States of America, **4** Department of Medicine, University of Washington, Seattle, Washington, United States of America, **5** Somali Health Board, Tukwila, Washington, United States of America, **6** Ethiopian Community Center in Seattle, Seattle, Washington, United States of America, **7** Ethiopian Health Coalition, Seattle, Washington, United States of America, **8** Eritrean Health Board, Seattle, Washington, United States of America

‡ SAH and NS authors contributed equally as first co-authors. RPK and RCP authors contributed equally as senior co-authors.
¤ Current address: BDB 811 | 1808 7th Avenue South, Birmingham, AL 35233
* sah2277@cumc.columbia.edu

## Abstract

Significant challenges promoting positive HIV testing behaviors among African immigrant communities in the U.S. persist, though existing community resources may be leveraged to improving these behaviors and increasing testing uptake. We conducted 30 key informant interviews and five focus group discussions (n = total 72 participants) among members of the Ethiopian, Somali, and Eritrean communities in Seattle, WA to identify these resources. Our findings highlight the following three main themes for responsive interventions: (1) capitalize on religious leaders and institutions as key facilitators; (2) leverage existing community resources, such as ethnic community centers, health boards, and healthcare professionals; and (3) utilize existing culturally-rich media for health promotion, centering on multi-linguality, -culturality, and -generationality. Our findings suggest that a wealth of community resources and resilience factors exist to leverage to improve HIV testing behaviors among African immigrant communities in the U.S.

**Data availability statement:** All relevant data are within the manuscript and its Supporting Information files.

**Funding:** This work was made possible by a National Institute of Allergy and Infectious Diseases (NIAID) Center for AIDS Research (CFAR) supplement award (P30 AI027757). The funders had no role in study design, data collection and analysis, decision to publish, or preparation of the manuscript.

**Competing interests:** The authors have declared that no competing interests exist.

## Introduction

The African immigrant community in the U.S. grew by 251% between the years 2000 and 2016. African immigrants in the U.S. are disproportionately affected by HIV. In 2019, the U.S. government unveiled the "Ending the HIV Epidemic (EHE): A Plan for America" initiative with the vision to reduce incident HIV in the country by at least 90% by the year 2030, based on the key pillars of diagnosis, treatment, prevention, and response [1]. The plan has designated 48 counties, two cities, and seven states as high priority areas for receipt of EHE funding and activities. The priority areas collectively accounted for over 50% of U.S. HIV diagnoses between 2016 and 2017 and include King County, where Seattle is located [2]. In King County, African immigrants now account for 10% of HIV diagnoses though they only make up 2% of the county's population [3–5]. Notably, experiences with HIV in the African immigrant community are unique [6,7]. In this population, HIV transmission tends to be primarily through heterosexual routes with infections acquired both locally and in countries of origin [8]. Additionally, members of this community are more likely to receive late HIV diagnoses compared to U.S. born persons in King County and nationally [5,9–11]. Though African immigrants are disproportionately represented in HIV diagnoses, members of these communities are likely to achieve later-stage HIV care continuum outcomes, such as viral suppression or retention in care, as good as or even better than those of their U.S. born counterparts [5,9,10]. Thus, early testing is often the major challenge in the HIV care cascade for African immigrant communities.

In an effort to offer early HIV testing, in a previous project, Harambee! 1.0, we worked with the Ethiopian, Somali, and Eritrean communities, the largest African immigrant communities in King County, to offer free, community-based preventative health services that included integrated HIV testing [6]. In our qualitative findings from this past work, HIV-related stigma emerged as the primary barrier to accessing HIV testing within these communities. While Harambee! 1.0 identified stigma as a barrier through direct service provision, the current project, Harambee! 2.0, extends this work by conducting deeper qualitative exploration of stigma mechanisms while simultaneously identifying community assets and resilience factors using an explicitly asset-based framework. Stigma reduction interventions in HIV prevention have been extensively studied through both implementation science and community-engaged research frameworks [12,13]. The Consolidated Framework for Implementation Research (CFIR) emphasizes the importance of understanding community context and leveraging existing social networks for sustainable intervention delivery [14]. Similarly, community-based participatory research principles highlight the value of asset-based approaches that build upon community strengths rather than focusing solely on deficits [15,16]. Our current qualitative work, conducted under Harambee! 2.0, was designed to further explore the impact of HIV-related and intersectional stigma on HIV testing access and behaviors. In this work, we also explored existing community resources or resilience factors that could be leveraged to address the upstream factors influencing HIV testing access and behaviors [17,18]. We explicitly approached this work to critically pivot away from "deficit" or "deficiency" centered work. Such an approach typically focuses on identifying and emphasizing the problems or

shortcomings within a community, often without considering the strengths, resources, and resilience factors that can be leveraged for change. This asset-based framework aligns with established community-engaged intervention models that prioritize community ownership and cultural assets as foundational elements for effective health promotion [15,16]. Our approach extends this literature by specifically examining how community resilience factors can be systematically identified and mobilized to address HIV-related stigma in African immigrant populations, a perspective that has been under-explored in existing implementation research. In contrast, our diverges from this perspective by focusing on the assets and existing community resources that can support and enhance health interventions, thus empowering the communities rather than viewing them solely through a lens of need or deficiency. Throughout our experience working with Ethiopian, Somali, and Eritrean communities, we have noted their sense of social cohesion and wealth of community resources [19].

## Methods

### Study setting, academic-community partnership, and positionality

This study was conducted by researchers at the University of Washington in partnership with leaders and health organizations from the local Ethiopian, Somali, and Eritrean communities. The research team was composed of the academic researchers, study staff, and community partners, with most individuals belonging to one of the three communities.

Given that our work involves individuals for whom English is not their first language, we use "preferred language" as the language an individual is most comfortable using, [20]. We consider "dominant language" to refer to English as the language that immigrants to the U.S. must assimilate into to satisfactorily navigate spaces in the U.S., such as healthcare settings [21]. Our use of "dominant language" conveys the power dynamics that are often at play for those who feel disadvantaged by English not being their preferred language. We avoid the use of "limited English proficiency" as it frames non-English preference from a deficit perspective. By making these distinctions we aim to capture these complexities of different languages, often used by the same individual in different settings, and the power structures that exist in a multilingual society.

### Data collection

Our overall data collection approach utilized key informant interviews (KIIs) and focus group discussions (FGDs). We chose to utilize KIIs to gather in-depth insights from key stakeholders prior to conducting FGDs. We conducted KIIs from October 2019 through January 2020, which allowed us to refine and adapt the FGD guides based on preliminary findings, ensuring relevance and depth in subsequent group discussions. FGDs were conducted from March through April 2020 and were held virtually due to the COVID-19 pandemic. This phased approach provided complementary data, with KIIs offering individual perspectives and FGDs capturing group dynamics and shared experiences.

### Sampling

Participants for both KIIs and FGDs were selected using purposeful sampling with input from community partners to determine recruitment methods and potential participants. To conduct KIIs we sought out healthcare professionals, persons living with HIV (PLWH), and religious and other leaders from East African immigrant communities. To recruit PLWH we used provider referrals from case management organizations and the UW-affiliated HIV clinic at the county hospital. Our recruitment period for the KIIs started on September 2, 2019, and continued until December 20, 2019.

Our initial KII data analysis showed that any HIV-related stigma reduction intervention should include religious leaders and institutions, leading us to heavily sample from these groups for the subsequent FGDs, but we also included other community leaders and members in FGD. We recruited participants for the FGDs over a two-month period, from January 1 to February 28, 2020. We did not attempt to intentionally recruit PLWH for the FGDs in order to minimize the possibility of inadvertent disclosure and/or traumatizing conversations. The demographic data collected was age, gender, country of birth, occupation, and religious affiliation. Sampling numbers were guided by theme saturation within participant categories.

## Interviewers and facilitators

KII interviewers and FGD facilitators were selected by community partners from their respective communities. Each selected interviewer/facilitator was bi- or trilingual (of two or more of the following languages: English, Amharic, Tigrinya, Somali, or Kiswahili) and was first generation American. Some interviewers had varying previous experience conducting interviews and all had significant prior experience conducting FGDs. Regardless of previous experience, all interviewers received centralized training in October of 2019 for the KIIs and February of 2020 for the FGDs.

## Ethics

Prior to both KIIs and FGDs, we collected oral informed consent and patient demographics, and a $50 cash reimbursement was provided. We provided participants with an oral consent script that outlined the key components of study participation. This script summarized the study's objectives, potential risks and benefits, and the study procedures. A study team member reviewed the script with the individual, ensuring their understanding and addressing any questions. All participants also received an informational sheet that provided a detailed description of the study purpose and summarized the consent process. Our informed consent forms were offered in several relevant languages (English, Amharic, Tigrinya, Somali, or Kiswahili). This verbal consent process was reviewed and approved by the University of Washington Institutional Review Board (IRB). The study was conducted according to the guidelines of the Declaration of Helsinki, and approved by the Institutional Review Board of the University of Washington (STUDY00003046, November 30, 2017)

## Interview/discussion procedures

Interviews were conducted in the participants' preferred language whenever feasible. PLWH participants were offered the option to be interviewed by project members not belonging to an African immigrant community due to concerns of potential disclosure [22]. Two participants opted to undergo interviews in this manner and their interviews and consents were conducted in English by the Latinx member of our study team. Facilitators led their respective FGDs, while notes were taken by another team member from the same community. FGDs were community-specific and conducted in the group's preferred language(s). FGDs for the Ethiopian and Somali communities were gender-specific, due to concerns regarding gendered power dynamics that may negative influence participation by women in mixed gender discussions, while the Eritrean community chose to conduct a mixed gender FGD. Three study team members identified joined the group at the start of each FGD to welcome and thank participants. These interactions were translated from the preferred language to English or vice-versa in real time by the FGD facilitator.

Depending on interviewee preference, KIIs took place in spaces such as ethnic community centers and health organization offices. The KIIs were conducted in several different locations throughout the metropolitan Seattle area. FGDs took place via Zoom. We recorded the KIIs and FGDs then translated and transcribed into English verbatim from the audio. The person who conducted the interview or led the group transcribed the recordings when possible. When transcription by the interviewer/facilitator was not possible, it was done by another member of the partnership team that belonged to the same community and then reviewed by the interviewer/facilitator for accuracy.

## Interview/discussion guides

The KII guide largely focused on deepening understandings of intersectional stigma around HIV testing, and covered the following five main domains: 1) interactions with U.S. healthcare system; 2) barriers to health screenings; 3) stigma around health screenings and HIV-related stigma; 4) intersectional stigma and its influence on HIV testing; and 5) how to reduce stigma around health screenings, including HIV testing. The FGD guide largely focused on selection of an intersectional stigma reduction intervention, and covered the following three main domains: 1) stigma around health

screenings and HIV-related stigma, including for HIV testing; 2) intersectional stigma and its influence on HIV testing; and 3) intervention development for stigma reduction around HIV, with emphasis on the role of religious leaders and institutions. To provide direction in selecting an intervention, three prototypes for HIV stigma reduction were discussed during the FGDs, with one of the interventions being faith based [23–26]. Our guide development was informed by the Earnshaw Stigma and HIV Disparities Model to use as a framework in designing probes for our interview guides [17,27]. The Earnshaw model, developed by Earnshaw and colleagues, highlights the role of community and individual level resilience resources as well as how various stigmas interact to impact structural and individual level processes. These various stigmas negatively impact HIV testing through multiple pathways, such as anticipated stigma surrounding a positive HIV test (due to HIV-related stigma and/or homonegativity) and reduced access to healthcare (anti-immigrant stigma manifested at a structural level). Additionally, Rao et al.'s multilevel stigma approach, [27] and the socioecological model, [28] also informed our guides and probes.

## Data analysis

We uploaded English transcripts to NVivo (version 12.0, QRS International Pty Ltd.). We utilized inductive coding methods. Three team members (SH, GB, NS) conducted the coding with guidance from (RCP, FM, and RPK). In a group setting, we developed an initial codebook guided by readings of the first few transcripts and the KII and FGD guides. As transcript coding progressed, we iteratively adapted the codebook to reflect new findings. The three coding team members coded the first transcript together, then they double coded two transcripts and met to resolve any discrepancies by consensus, and then coded individually with SH reviewing all transcripts' coding. To systematically capture divergent perspectives, we used NVivo's comparison queries to examine differences between KII and FGD responses, as well as between PLWH and other participant categories. We maintained detailed analytic memos throughout the coding process to document patterns of convergence and divergence across participant groups and data collection methods. We held several group meetings to conduct thematic analysis, [29,30] and identified three overarching thematic domains that encompassed our emerging parent and child codes. These codes were then rearranged into the thematic domains with corresponding convergent and divergent subthemes accompanied by illustrative quotations.

## Results

This study included 72 total participants from 30 KIIs and five FGDs, with one participant participating in both a KII and an Ethiopian FGD. Of the total sample, 27 (37%) of participants were born in Somalia, 26 (36%) were born in Ethiopia, and 17 (23%) in Eritrea. All participants were born in the country whose community they identified with, with the exception of one U.S.-born participant who identified with the Eritrean community and one Kenyan-born participant. Participants' age ranged from 22–67 years of age, however, majority were in the range of 30–49 years. Five study participants who participated in the KIIs identified as PLWH and nearly half (n = 35) identified as women. A majority of participants reported working as either religious leaders (n = 14) or healthcare professionals (n = 15). Table 1 provides further details on the sociodemographic characteristics of the participants.

Three major themes emerged from our work to identify community resources and resilience factors to leverage for change in HIV testing behaviors among the Ethiopian, Somali, and Eritrean immigrant communities in the Seattle area,: 1) the role of religious leaders and religious institutions as key facilitators for access to and influence of community members, 2) existing community resources, including ethnic community centers, health boards, and healthcare professionals, and 3) reliance on ethnic media outlets and social media platforms. We present these three major themes in further detail below. In general, there were far more commonalities across the three communities and between the KIIs and FGDs than differences. Thus, when differences exist, we highlight the differences, otherwise the findings are applicable to all three country-of-origin communities queried in this study. Table 2 provides example quotations from the interviews and focus groups, organized by themes as described below.

**Table 1. Sociodemographic characteristics of study participants, n (%).**

| | Total participants | Key informant interviews (KIIs) | Focus group discussions (FGDs) |
|---|---|---|---|
| | (n = 72)* | (n = 30) | (n = 43) |
| **Age** | | | |
| <30 | 5 (7%) | 2 (7%) | 3 (7%) |
| 30-49 | 48 (66%) | 16 (53%) | 32 (74%) |
| 50+ | 20 (27%) | 12 (40%) | 8 (19%) |
| **Gender** | | | |
| Male | 38 (52%) | 16 (53%) | 22 (51%) |
| Female | 35 (48%) | 14 (47%) | 21 (49%) |
| **Country of birth** | | | |
| Ethiopia | 27 (37%) | 11 (37%) | 16 (37%) |
| Eritrea | 17 (23%) | 8 (27%) | 9 (21%) |
| Somalia | 27 (37%) | 9 (30%) | 18 (42%) |
| Kenya | 1 (1%) | 1 (3%) | 0 |
| U.S. | 1 (1%) | 1 (3%) | 0 |
| **Language of interview** | | | |
| Amharic | 25 (34%) | 9 (30%) | 16 (37%) |
| Somali | 27 (37%) | 9 (30%) | 18 (42%) |
| Tigrinya | 18 (25%) | 9 (30%) | 9 (21%) |
| English | 2 (3%) | 2 (7%) | 0 |
| Kiswahili | 1 (1%) | 1 (3%) | 0 |
| **Occupation** | | | |
| Healthcare professional | 15 (21%) | 8 (27%) | 7 (16%) |
| Religious leader | 14 (19%) | 4 (13%) | 10 (23%) |
| Business/management | 11 (15%) | 5 (17%) | 6 (14%) |
| Education/student | 10 (14%) | 5 (17) | 5 12%) |
| Homemaker | 6 (8%) | 1 (3%) | 5 (12%) |
| Laborer | 3 (4%) | 1 (3%) | 2 (5%) |
| Not Working | 3 (4%) | 2 (7%) | 1 (2%) |
| Other community leader | 2 (3%) | 2 (7%) | 0 |
| Other | 9 (12%) | 2 (7%) | 7 (15%) |
| **Community** | | | |
| Ethiopian | 27 (37%) | 11 (37%) | 16 (37%) |
| Eritrean | 18 (25%) | 9 (30%) | 9 (21%) |
| Somali | 27 (37%) | 9 (30%) | 18 (42%) |
| Kenyan | 1 (1%) | 1 (3%) | 0 |
| **PLWH** | | | |
| Yes | 5 (7%) | 5 (17%) | N/A |
| No | 25 (34%) | 25 (83%) | N/A |
| **Religious Affiliation** | | | |
| Orthodox Christian | 12 (16%) | N/A | 12 (28%) |
| Evangelical Christian | 9 (12%) | N/A | 9 (21%) |
| Islam | 20 (27%) | N/A | 20 (47%) |
| Catholic | 1 (1%) | N/A | 1 (2%) |
| Protestant | 1 (1%) | N/A | 1 (2)% |

*One individual from the Ethiopian community participated in both a KII and a FGD.

**Table 2. Main themes, subthemes, and example quotes for community resource and resilience factors among Ethiopian, Somali, and Eritrean immigrant communities in Seattle, WA.**

| Main theme | Subtheme | Example quotes |
|---|---|---|
| **(1) Religious leaders and institutions as key facilitators** | | "The elders and religious leaders would be the best people to be trained and given this information in order to create awareness within the community." *(56 year old male, Somali, imam/religious leader)*<br>"Reaching people during church sermons might help as well. It should be delivered in a simple way. There are issues of literacy in our community. So the teachings and information should be delivered in a simple and easy to understand way to get people's attention." *(34 year old female, Ethiopian, nurse)*<br>"And as I said earlier, working with community centers, churches, mosques and other religious institutions, inviting the leaders of these institutions and organizing workshops and seminars, and providing incentives. Moreover, providing free screening with collaboration with these institutions and follow up" *(40 year old female, Eritrean, housewife)*<br>"It will be ideal if they agree to take 5 minutes every Sunday after church service to talk about awareness and provide information to members. It must be done uniformly so the message will reach everyone in the community. We need to translate the message in different languages and present the information at different events like health fairs." (Ethiopian, FGD 1)<br>"Homosexuality is a major issue in the community so there is no doubt that they would stigmatized. If someone is homosexual, they would never risk to be recognized so they would never do HIV screening." *(40 year old female, Eritrean, housewife)* |
| | A) Religious leaders and institutions incorporating health education and promotion in their teachings | "Religious leaders can play a very big role and we always say that. We say that this disease or infection is a wrath from God. So, the religious leader suffered very big role to play within society. The role of the legislators normally within the society is to treat society of psychological problems in understanding everyday life situations. If they see someone struggling to understand in parts of the faith, the religious leaders need to step in and help shed some light on some of the issues. Previously, they often spoke on issues of family and children, other safety issues within the community. The same way we talked to the community about all these other issues, we can talk to them about them, HIV and AIDS as well." *(56 year old male, Somali, imam/religious leader)*<br>"Another thing I want to add, right now the Ethiopian community have created a task force made up of spiritual leaders from the orthodox church, evangelical church, Mosque, and other faith-based institution to tackle mental health issues in our community. Educating these individuals is essential because most believers or members listen to what their spiritual leaders tell them. Each leader has large number of followers and communicating our message through them will be an excellent idea. People trust their leaders, and for certain groups of populations who are not familiar with social media or do not have Facebook, using their spiritual leaders to communicate any messages will be very productive." (Ethiopian focus group #1)<br>"It [HIV] is not a life sentence. You can go to churches also the congregation it is good for them to be educated about it even if it's a five minute talk. It will help us all in that church, you can get one or two. Yeah, it will help somebody. At least someone will come out and someone maybe has been dying with it and they don't know where they can reach out. So you don't know how many souls you'll reach out in the community. Even in the church, you never know." *(38 year old female, Kenya, banker)*<br>"You can organize a symposium by the health professionals in any meeting hall. Second, you invite religious leaders and community leaders. You distribute flyers in religious and community centers. Health professionals, nurses and doctors especially HIV/AIDS specialists; and social workers and religious leaders would execute it. You use both religious and social venues to reach out to the community to ensure participation. This can be uniting, too" *(50 year old male, Eritrean, nurse)*<br>"Religious institutions and other organizations in our community have not done anything so far to reduce the stigma associated with HIV. They also have not done much to teach the community about the virus. It is very disappointing. More than the community center I lay the blame on religious institutions because that is where the majority of the population largely congregate." *(54 year old male, Ethiopian, retiree)* |

*(Continued)*

**Table 2.** (Continued)

| Main theme | Subtheme | Example quotes |
|---|---|---|
| (2) Existing healthcare professionals and resources in the community | A) Confidence and connection community members have with ethnic organizations as well as the comfort in approaching them with health concerns | "If you say to the community "a doctor will come and teach you" nobody will come, however, if you start organizing and creating awareness about it using people from their own community, the doctor can come and make an impact because they are organized and aware about it. So, start from the grass root using their own people and then you can proceed to the next level." *(38 year old male, Eritrean, supervisor)* <br><br> "I think many members of the community are very connected with the community organizations such as the Somali Health Board. That is where most of the community members come with any questions that they may have about health. In my opinion, I believe the most appropriate organization that would do some sort of creating awareness on issues relating to health, would be some health board. So, when the information is coming from organizations that the community understands and trusts, in like some health board, I think after a while everybody would learn about this and the issues will be normalized again. The change will come if the right education and awareness created." *(40 year old female, Somali, community health worker)* <br><br> "The religious community is ready now. It's up to the health professionals and boys and girls from the community who aren't health professionals to come together and create this awareness. It would really be nice if they came together one of those days in a community gathering and bring some food for people to eat and talk about this openly. That would be very nice for the community." *(56 year old male, Somali, imam/religious leader)* <br><br> "Health professionals, Eritrean professionals, community leaders, religious leaders, especially I would involve young people in the program/project." (*38 year old male, Eritrean, supervisor*) <br><br> "Another issue might be the current political situation in Ethiopia which has a spillover effect here. People are divided by ethnic group and political views. Most people do not go to the community. These days, only certain groups go to the Ethiopian community center because of the division that exists. For example, there is Gondor community, Tigray community … and these are not part of the Ethiopian community. There is a lot of division which makes it hard to reach the maximum number of people easily. Maybe using churches and similar forums might be a good idea to reach many community members. Otherwise, it is hard to reach them. <br> But going back to the question of who should lead the project, I think maybe having someone not from the community to lead the project might be important. Our community is very difficult. They might take it more seriously if someone not from the community is leading it. Especially because the community is divided based on ethnicity, religion and other things, if a member of one community leads this project the others might not be interested to work together or might not take it seriously or trust it at all. If an independent person leads the project, it might give the issue the necessary weight. For example, someone from Seattle mayor's office should lead this project." *(34 year old female, Ethiopian, nurse)* |
| | B) Using existing, trusted health professionals from the community | "This would need existing community support systems like the Somali Health Board to exert even greater effort in helping provide the support that all the community members need, especially the ones who are facing discrimination and stigma due to the illness. Individuals who are facing discrimination and stigma may hopefully find the comfort and the support they need in difficult times." (Somali focus group #2) <br><br> "I would like to add and emphasize, the community has a great need when it comes to health resources. In addition to the great resources that organizations like the Somali Health Board are already giving the community, there needs to be even greater outreach and community wide education and engagements within the community and in places where the community gathers." *(40 year old female, Somali, community health worker)* |
| | C) Annual health fairs hosted by ethnic health boards or committees aid | "Since the organization called the Somali Health Board has been created, the community is seeing its benefits. Everybody who is working there is the same community members that they can trust. So they do annual health fairs in which there are health screenings and they invite so many health providers to that health fair. More often, some people who have never had health screenings and when they come to the health fairs, they find out that they are in critical condition and they need help right away. And they connect them to the health providers and many of them get help and some of them actually get hospitalized because of how grave their condition is. So why are they more forthcoming in this kind of setting than when they are with their doctors, the simple answer is trust. Because they believe that these people are looking out for their best interest, and the best interest and welfare of the community unlike the doctors in the health system. So, somebody's health would have changed the perception of how health care is perceived throughout the community." (*58 year old male, Somali, social worker/community activist*) |

*(Continued)*

**Table 2.** (Continued)

| Main theme | Subtheme | Example quotes |
|---|---|---|
| (3) Culturally-rich resources; such as existing media outlets, and community ties | A) Reliance on ethnic media outlets/ social media platforms | "One more thing I would like to add, we need to create a strategy to allocate one month to talk to specific community and faith leaders and work on training them on how to communicate HIV prevention education. The next month we can dedicate the entire month to talk about prevention, the next month, we can talk about treatment. I think doing community outreach and promoting prevention in this manner will eliminate information overload and confusion. I think if we collaborate with everyone and manage to do it at once, we will be more effective." (Ethiopian focus group #1)<br>"Using the community centers, churches, youth clubs/groups. Setting up some kind of system like a website or hotline so that individuals can use it to get information while keeping their privacy would help. It also can be educational videos that anyone can watch privately and learn or get the necessary information. For example, using radio programs and TV channels might be helpful. And if there is one website with all the necessary information, like about these programs and where to get help I think it will be very helpful. This way everyone can get all the information needed on one website and also keep their privacy. And we should publicize this website through different means like through the community center, churches and other means." *(58 year old male, Ethiopian, pastor)*<br>"While social media platforms like Snapchat, WhatsApp, Facebook are very important platforms, there also need to be continuous community engagements and face-to-face meetings of at least twice a month to share information. People and the community need to understand that the existence of these diseases in the proper professions and treatments." *(40 year old female, Somali, community health worker)*<br>"Youth should be the targeted audience" *(62 year old male, Eritrean, manager)* |
| | B) Incorporating cultural parables and word-of-mouth techniques to relay health information | "I don't choose video as people are overwhelmed with social media and YouTube. People are tired of it. There is sweetness when a person comes and teaches you in-person. If you come in-person and teach me I may hear you with open heart and any mistakes I have done can be corrected with understanding." *(38 year old male, Eritrean, supervisor)*<br>"We're very verbal oral community. It's good to make a flyer or a poster or whatever that is so somebody has something to go home with. But unless you're sitting and having a conversation and it's a verbal exchange that information will never be fully taken in by the community." *(32 year old female, Somali, doctor)*<br>"I may use spiritually, and culturally based education. In our community we do have way of increasing people's awareness. Particularly, parables are effective in educating our community. There are stories in our culture about how people help each other, which is important to incorporate within the education." *(62 year old male, Eritrean, manager)*<br>"We can use music, spiritual song, powerful words can get into people easily and govern. As I told you, there are powerful preachers. For example, even in mosque, there are people who teach the Quran very well. Following the lesson by professionals, the class has to be given on daily basis. Unless there is no continuity, it will stop as is the case with classroom lessons. Even if generations come and go the education has to continue." *(55 year old female, Ethiopian, barista)* |
| | C) Importance of centering messages with preferred languages | "I think it is something that prevents a lot of people from accessing health care in the way that they would want to. There are huge cultural issues. There's a huge intimidation accessing a space when people don't look like you or don't speak your language and you are a burden. So, there is a lot of that, that continues until today and with each community, they face unique challenges and the system like I said still has institutionalized and structural racism. Which makes it difficult for people who have intersectional identities to access." *(32-year-old female, Somali, doctor)*<br>"There are issues of literacy in our community. So the teachings and information should be delivered in a simple and easy to understand way to get people's attention." *(34 year old female, Ethiopian, nurse)*<br>"I think the best audience would be all ages but more specifically parents that English is their second language in a sense that, people that come from… I am not saying that people that were born here do not need that education or that awareness, I think it is more critical to focus or put more energy for those who come from Eritrea or other communities because it is hard to know what kind of experience and knowledge they have about that specific disease and just like doing activities that makes it normal to get tested for HIV or to have a regular breast cancer screening or colon cancer screening. It is something we should do on a regular basis like we do everything else." *(38 year old male, Eritrean, supervisor)* |
| | D) Leveraging strong community bonds | "There are very minimal degrees of separation around our community members. Everyone knows everyone. So, once the diagnosis like that (HIV) gets out, that person is no longer feels, at least to me, no longer feels welcome in the community. Because everyone knows or everyone has talked about it…so no one wants to share a cup of tea. no one wants to share a bathroom. Things like that, I think are still happening in the community." *(32 year old female, Somali, doctor)*<br>"Even though it could be a big city, everyone is usually very connected to their own community. If someone is diagnosed with HIV and AIDs for example, they will face a widespread backbiting and other name calling stigmas. If the person were married it is possible that their wife would leave them. And society might cut them off completely with no interactions at all." *(38 year old male, Somali, student)* |
| | E) Learning from PLWHIV | "Educating the community on this would be helpful. This can be done through fliers and other ways. For instance, it can be done by using HIV positive people who are willing to talk about it in public. If these people are willing to share their experience, I think it is will be a huge benefit. There is one individual that comes from Minnesota, he is HIV-positive and he goes out in public and talks about HIV. He was very famous in Ethiopia and now he has moved to Minnesota. When he comes here at community meetings, he talks about his status openly. So, this kind of approach might be beneficial. And I think repeated information and awareness raising events should be held. It shouldn't be once in a while but a repeated event. It Is also important to teach community leaders on how to treat HIV positive individuals." *(58 year old male, Ethiopian, pastor)* |

## 1) Religious leaders and institutions as key facilitators

Across all three ethnic communities, both KII and FGD participants reported that religious leaders and institutions serve as important facilitators for their communities. First, participants in both KIIs and FGDs noted that these leaders and institutions exert considerable influence on their communities, with KIIs more frequently emphasizing their potential to mitigate stigma, while FGDs highlighted concerns about their role in perpetuating it. Participants indicated that the immense trust community members have in religious leaders makes them crucial allies to collaborate with for positive change. Participants also indicated that most members in their communities belong to religious institutions and those institutions play an instrumental role in their lives, from shaping their worldviews to influencing their day-to-day activities.

*The elders and religious leaders would be the best people to be trained and given this information in order to create awareness within the community.*"(56 year old male, Somali, imam/religious leader)

As conceptualizations of disease or illness are often tied into religious or spiritual beliefs, it is only appropriate that the religious leaders and institutions themselves incorporate health education and promotion in their teachings. Religious leaders, with their positionality and influence within these communities, appear to be the natural community leaders to respond to conceptualizations of disease, given that community members often mention disease as a punishment from God or the result of fate. In addition, participants from both the KIIs and FGDs identified homosexuality as a highly stigmatized identity within the community, with the majority of religious leaders and institutions in these communities upholding negative beliefs about homosexuality, morality, as well as its associations with HIV. It was mentioned that this homonegativity deters people from testing for HIV, due to the assumption that testing is only needed for those engaging in homosexual sex. As a result, most people choose to avoid learning their status rather than risk being stigmatized for same-sex behavior or perceived religious transgressions.. These homophobic beliefs are debilitating to the point where participants acknowledged that families would choose to hide the HIV status of their loved ones until they die, rather than disclose the positive status openly with their community. We also observed that at times that attributing immorality to people living with HIV may be independent of any homonegativity. Many participants thought their community members believed an HIV diagnosis as a curse on their family, implying a religious connotation, regardless of sexual orientation. One participant even stated that HIV is considered the "devil's" disease. Given all the associations between HIV and religious beliefs, our participants felt strongly that involving religious leaders and institutions was paramount in any work to address HIV. Participants from both KIIs and FGDs suggested concrete ways in which religious leaders or institutions could influence community beliefs around HIV. For example, some participants noted that including health education or promotion in sermons is a powerful tool in mitigating stigma around HIV, and one participant specifically noted that having these leaders stress that seeking medical care does not adversely affect their relationship with God as an important message to convey during sermons. Therefore, empowering religious leaders to positively message HIV can effectively reach many community members, and working on ways to appropriately educate the religious leaders should be a priority item for work addressing HIV in these communities.

*"Homosexuality is a major issue in the community so there is no doubt that they would stigmatized. If someone is homosexual, they would never risk to be recognized so they would never do HIV screening." (40 year old female, Eritrean, housewife)*

*"It [HIV] is not a life sentence. You can go to churches also the congregation it is good for them to be educated about it even if it's a five minute talk. It will help us all in that church, you can get one or two. Yeah, it will help somebody. At least someone will come out and someone maybe has been dying with it and they don't know where they can reach out. So you don't know how many souls you'll reach out in the community. Even in the church, you never know." (38 year old female, Kenya, banker)*

## 2) Existing wealth of community resources, including ethnic community centers, health boards, and healthcare professionals

Participants identified a wealth of community resources already available in their communities to be leveraged for positive change. Across both FGDs and KIIs, participants repeatedly stressed the interconnected nature of social networks or strong social cohesion among the Ethiopian, Somali, and Eritrean communities in the U.S. This social cohesiveness is driven by ethnic, sometimes even tribal-level, and country-specific identities. They highlighted that, while social cohesion can be detrimental to HIV testing (as we have described at length elsewhere) [31], social cohesion can also be leveraged for positive change. One key element of social cohesion our participants identified was community members coming together, facilitated in one form or another by trusted leaders and institutions.

*"Religious institutions and other organizations in our community have not done anything so far to reduce the stigma associated with HIV. They also have not done much to teach the community about the virus. It is very disappointing. More than the community center I lay the blame on religious institutions because that is where the majority of the population largely congregate." (54 year old male, Ethiopian, retiree)*

FGD participants, particularly community members without a known HIV diagnosis, affirmed the confidence and connection community members place in ethnic organizations and their willingness to seek health-related guidance from them. The Ethiopian, Somali, and Eritrean Health Boards emerged as key resources in advocating for and educating their communities to effectively respond to and raise awareness regarding HIV.

PLWHIV interviewed in KIIs strongly emphasized that health education on stigmatized illnesses, including HIV, must come from within the community rather than from external sources. FGD participants also echoed this sentiment, stressing that utilizing trusted health professionals from the community to provide education on stigmatized illnesses is far more effective in driving change than bringing in someone who isn't a member of the community. They indicated that their community members want to hear from others from their own communities, including existing health professionals. To have the conversations led by health professionals from the community would also encourage a more "grassroots" approach, which participants felt would be better perceived by their community members. One participant even mentioned HIV education should only be done by health professionals who are from and live in the community, stressing that trusted health professionals from the community can provide culturally sensitive appropriate education.

*"If you say to the community "a doctor will come and teach you" nobody will come, however, if you start organizing and creating awareness about it using people from their own community, the doctor can come and make an impact because they are organized and aware about it. So, start from the grass root using their own people and then you can proceed to the next level." (38 year old male, Eritrean, supervisor)*

Both KIIs and FGDs highlighted the role of annual health fairs hosted by ethnic health boards or committees aid community members who have not previously engaged in preventive health care and promote engagement by connecting participants to health providers from their own communities in a culturally secure environment or context. Healthcare professionals from their own communities are valued resources, because they are perceived to facilitate a sense of comfort and faith in this process given their cultural and linguistic understanding of the communities. Some participants felt that the health professionals from their communities may also be better able to reach youth rather than professionals from outside their communities. The idea of meeting the community where they are with health services was thought to be crucial in making these services more accessible, as opposed to promoting preventive health care activities that asked the community members to engage in existing health systems, such as doctor's offices.

**PLWH as trusted community educators.** A distinct subtheme that emerged was the unique role that PLWH could play as community educators. Participants were eager to have PLWH from their community share their lived experiences with other community members, feeling that knowing someone in their community living successfully with HIV would help their community members think in a very socially-connected way about HIV. Some did acknowledge, however, that the high stigma around HIV would make it hard for many PLWH in their community to come forth and disclose their status, yet several participants did indicate that one or two persons in the various East African communities who they knew had widely disclosed their positive HIV status. Having those individuals who are comfortable with disclosing their status and sharing their experiences would have tremendous meaning for community members. This perspective was particularly emphasized by PLWH participants in KIIs, who stressed the authenticity and credibility that comes from lived experience within these tight-knit communities.

### 3) Culturally-rich, existing ethnic media resources, including social media

Our findings showed that the utilization of multifaceted and existing communication channels are crucial to maximizing the benefits of trusted and well-known resources within the community. Awareness of these services are widespread among community members. Participants suggested that short programs on ethnic television channels specific to each community might effectively change perceptions around HIV. Social media, many felt, could also be leveraged as an additional platform for awareness. Importantly, participants did note that generational differences are apparent in the reach of each medium type, since elderly members often prefer in-person outreach while the youth are very engaged with social media. Regardless of the medium, participants stressed that a personal connection with the source of the information is of prime importance given the strong social cohesiveness and networks in their communities. Our participants also highlighted that verbal communication of health information is key for "word-of-mouth" communities.

*"While social media platforms like Snapchat, WhatsApp, Facebook are very important platforms, there also need to be continuous community engagements and face-to-face meetings of at least twice a month to share information. People and the community need to understand that the existence of these diseases in the proper professions and treatments."* (40 year old female, Somali, community health worker)

*We're very verbal oral community. It's always good to make a flyer or a poster or whatever that is so somebody has something to go home with. But unless you're sitting and having a conversation and it's a verbal exchange that information will never be fully taken in by the community."* (32 year old female, Somali, doctor)

Additional elements highlighted by participants included a variety of suggestions related to more effectively targeting messaging around HIV for their communities. First, some participants were eager to have PLWH from their community share their lived experiences with other community members, feeling that knowing someone in their community living successfully with HIV would help their community members think in a very socially-connected way about HIV. Some did acknowledge, however, that the high stigma around HIV would make it hard for many PLWH in their community to come forth and disclose their status, yet several participants did indicate that one or two persons in the various East African communities who they knew had widely disclosed their positive HIV status. Having those individuals who are comfortable with disclosing their status and sharing their experiences would have tremendous meaning for community members. Second, some participants thought that incorporating parables of community members supporting each other within health education messages would resonate well with their communities, again highlighting a layer of social connectedness being a strength to foster in health promotion. Third, another suggestion was to facilitate informal discussions of HIV through community gatherings where members can congregate and eat together, an important element to how these communities socialize. Many of the existing ethnic health boards or community centers already organize such events for their community members, for example, via a monthly breakfast for their elders; participants thought such opportunities should

be leveraged to help positively message HIV. Fourth, the importance of centering messages with appropriate languages was noted, as participants felt that messaging in English for communities whose preferred languages are not English is a barrier to effective health promotion.

**Summary of key convergences and divergences.** Across all three communities and participant types, there was strong convergence on the importance of community-led approaches, leveraging social cohesion as a strength, and the need for culturally and linguistically appropriate messaging. All participants emphasized that interventions must come from within the community rather than external sources. However, notable divergences emerged between participant types and data collection methods. KII participants, particularly PLWH, placed stronger emphasis on community-led education and the unique credibility of lived experience, while FGD participants focused more on institutional and organizational approaches. Religious leaders in KIIs emphasized their potential for positive messaging, while FGD participants expressed more concern about religious institutions perpetuating stigma. Additionally, clear generational differences emerged regarding communication preferences, with older community members favoring in-person approaches and younger members preferring social media platforms. These divergences highlight the importance of multi-faceted intervention approaches that can accommodate different perspectives and preferences within these communities.

## Discussion

Our study identified several overarching themes for existing community resources or resilience factors that can be leveraged to decrease HIV stigma and improve HIV testing behaviors among the Ethiopian, Somali, and Eritrean immigrant communities in King County, WA. These findings contribute to multilevel stigma reduction frameworks by demonstrating how community assets can be systematically mobilized across individual, interpersonal, and structural levels to address HIV-related stigma. First, religious leaders can play a vital role in re-guiding beliefs surrounding HIV and morality, and religious institutions are a place where stigma-perpetuating practices can be interrupted to instead decrease HIV stigma. Second, a wealth of existing community resources, including the community centers, ethnic health boards, and health professionals, can be leveraged to lead design and implementation of health education or promotion. Lastly, utilizing the multifaceted and existing, culturally-rich ethnic media outlets, including ethnic broadcast stations, social media, and in-person health promotion talks, can build on existing trust and comfort with these outlets while simultaneously reinforcing social-connectedness for health promotion. Existing resources that are already meaningful for these communities should be prioritized in any efforts to decrease HIV stigma within these communities.

### HIV education in religious organizations

Our findings advance understanding of how religious institutions can serve as implementation sites for stigma reduction interventions, addressing a critical gap in the Earnshaw Stigma and HIV Disparities Model regarding faith-based community resilience resources. Religious leaders hold remarkable positions of power and trust in these communities, so empowering religious leaders to address HIV stigma could be a powerful tool to address HIV in these communities. Our study found that shame and stigma associated with HIV can be debilitating to the point where family members feel the need to hide a family member living with HIV from the public [32,33]. Some PLWH decline participation in religious organizations, primarily due to a fear of involuntary disclosure and the associated stigma [34,35]. A clear opportunity to promote HIV education and awareness via religious systems exists, which some projects have actively done in the U.S [36, 37].

Another challenge our participants anticipated in working with religious leaders is disassociating the link between homonegativity or "immoral" sexual practices and HIV within these communities. Yet, we repeatedly heard from our participants and community partners that religious leaders and institutions are crucial contenders to collaborate with for positive change. We believe partnering with community organizations who already have trusting relationships with religious leaders and institutions can be a way forward. For example, one of our partners is already working with religious leaders to promote health education regarding diabetes care, a disease less stigmatized than HIV in these communities

[19]. This partnership is now ready to include addressing HIV stigma. Another way forward, as we have found in our prior work, [6] can be to integrate HIV education into a broader agenda to address various health diseases. Lastly, given the layers of discrimination these communities face due to their intersecting identities [38], applying a health equity and social justice lens to health promotion activities, particularly in religious spaces, could be a powerful strategy [39]. This approach could help address not only health disparities but also the broader social determinants of health that disproportionately affect these populations [39]. By integrating health equity and social justice principles, religious leaders and community members could advocate more effectively for systemic changes that promote equitable access to healthcare and reduce stigma [39]. This framework empowers these communities to challenge structural barriers, advance health literacy, and create more inclusive health policies, ultimately fostering a more supportive environment for health promotion.

## Utilizing existing community resources

Our asset-based approach directly challenges deficit-focused models in implementation science by demonstrating how community pride and existing social capital can serve as intervention foundations. Work among Boston's immigrant Somali community found that a robust social support network was crucial for sustaining a resilient community, that the strength of the social cohesion was a point of pride [40]. Therefore, the challenge for those working to infuse behavior change in these communities is to understand how to tap into those tight-knit social networks for positive change. For instance, during the COVID-19 pandemic, community partnerships have successfully communicated emergency information to vulnerable populations by leveraging existing networks [41]. Furthermore, utilizing existing community information sharing networks could be instrumental in overcoming the barriers of mistrust of the U.S. healthcare system, a key barrier in accessing HIV testing and other preventative healthcare services [6]. Likewise, other barriers such as language preferences and intersectional stigmas would be easier to overcome by leveraging the trusted channels and leadership already established by the communities themselves [42].

## Contribution to implementation science and stigma reduction literature

Our findings make several key contributions to implementation science and multilevel stigma reduction frameworks. First, we extend the Earnshaw Stigma and HIV Disparities Model by providing concrete examples of how community and individual resilience resources can be systematically identified and mobilized within immigrant populations [17,18,43]. Our asset-based approach demonstrates how the "community resilience" component of this model can be operationalized through religious institutions, health boards, and media networks. Second, our work advances implementation science by illustrating how the Research CFIR constructs of "networks and communications" and "cosmopolitanism" can be leveraged within immigrant communities [14,44]. The transferability of our approach suggests that other immigrant groups facing similar challenges, including language barriers, healthcare mistrust, and intersectional stigma, could benefit from parallel asset-identification processes. This framework could be adapted for other African immigrant populations by focusing on their unique cultural strengths while applying similar principles of community-led intervention development.

## Developing and implementing an intervention

We found that all three communities suggested utilizing word of mouth, especially for the elderly, to effectively communicate health information in these communities, which often have rich oral history traditions [45]. For example, parables that embody connotations of religion can be leveraged for health promotion, in ways that are more acceptable, and even comfortable, for the community members. Additionally, our findings highlight perceived challenges in reaching different generations with HIV prevention information. Social media, with its impersonal and private use, and other online resources were often noted as being easier to navigate by younger vs. older generations, and that elderly members of the communities preferred in-person outreach. Thus, enabling health promotion or education that uses oral stories or informal communication via word of mouth may be influential in decreasing HIV stigma. This has been done previously to promote HIV testing

and condom use among a Latinx community, where "traditional" communication in person performed better for changing attitudes towards HIV testing than social media [46]. Nonetheless, we acknowledge that implementing interventions promoting informal communication are fraught with the many inherent difficulties in developing tools that can be standardized and ensure fidelity for intended information.

Relatedly, we encountered concerns regarding older individuals' difficulty in using English as the dominant language and feeling disempowered to not use their preferred language when navigating U.S. healthcare systems. While non-English language interpretation is frequently available in-person or telephonically, these services have many limitations, from inaccuracies in interpretation to concerns over loss of confidentiality [19]. Thus, community-led, in-person outreach may help center the intervention on preferred languages and foster accessibility and reach. Accessing information and healthcare in an individual's preferred language is not only vital for understanding but also has the potential to mitigate other barriers such as medical mistrust. Thus, interventions for these communities must include elements that prioritize multilingual and multigenerational, as well as culturally-centered programs.

Furthermore, while the research was conducted in partnerships with only three East African communities, the shared cultural, social, and health-related experiences among them suggest that the insights gained could have broader applicability with nuances accounting for specific emphasis or highlights that may differ. The commonalities observed in the ways these communities approach health, particularly around stigma and access to care, may resonate with other African immigrant populations facing similar barriers. Additionally, immigrant groups from different regions who experience parallel challenges related to acculturation, health disparities, and access to services may benefit from the strategies and interventions developed from this study. This potential transferability underscores the importance of understanding community-specific factors while recognizing broader systemic issues that influence health behaviors and outcomes in immigrant populations. Further research is needed to explore how these findings can inform health interventions for a wider array of immigrant groups.

## Limitations

Our study has several limitations. First, we acknowledge that our approach to understanding community resources explicitly pivots away from focusing on any deficiencies in resources, so we do not aim for "comprehensiveness." Second, this study was conducted in exclusive partnership with the Ethiopian, Somali, and Eritrean communities in King County. However, the commonalities between the three groups suggest that transferability to other African immigrant communities or perhaps other immigrant groups is possible. Furthermore, relying on community partners for recruitment introduces potential bias. However, their engagement with community leaders fostered trust and confidence. Generic strategies like flyers or social media are less effective because these communities rely more on interpersonal connections. Future efforts can incorporate diverse recruitment methods to reduce bias. Third, KII and FGD participants were selected to represent target subgroups, such as PLWH and religious leaders, which may have reduced the breadth of community perspectives included in the study, though others still participated, resulting in an adequately diverse range of community perspectives. Fourth, potential social desirability bias may have influenced responses, particularly in community-based FGDs discussing a highly stigmatized topic like HIV, where participants may have been reluctant to express views that could be perceived negatively by peers or community members. Fifth, the COVID-19 pandemic introduced unique challenges to the study, particularly in transitioning to virtual focus groups. While KIIs were conducted in person, the pandemic required the FGDs to be moved online. This transition to virtual focus groups also required participants to have a certain level of digital literacy and comfort with technology, which may have excluded some individuals from taking part. Despite these challenges, the research team adapted well to virtual formats, and the data collected from both KIIs and FGDs remain valuable in capturing participants' perspectives. The shift to virtual FGDs may have impacted the dynamics of group interaction, but our overall findings were consistent across both methods. Despite these limitations, our study has conducted unique work with the Ethiopian, Somali, and Eritrean U.S. communities to develop a greater understanding of community resilience resources as they relate to HIV stigma.

## Conclusions

In conclusion, our findings show the significant challenges of dismantling HIV stigma and promoting positive HIV testing behaviors which can be addressed by leveraging the plethora of existing community resources and resilience factors in the Ethiopian, Somali and Eritrean communities. Responsive interventions must utilize the strong social cohesion and networks that already exist within these groups. These include community leaders, such as religious leaders and health professionals, as well as physical and virtual spaces for information sharing and community building. Mitigation of HIV stigma starts with education through already established and trusted resources, whether it be religious or community-oriented spaces such as mosques/churches, ethnic community centers, or ethnic health boards. Additionally, health promotion in these communities must center on multi-linguality, -culturality, and -generationality to better reach the various members of their communities. Finally, our findings show that proper facilitation of conversations surrounding HIV and larger health inequities can be a starting point in encouraging vulnerable communities to take the next step towards advocating for themselves. Building on these findings, our next steps involve developing and piloting a community-led intervention in partnership with the same Ethiopian, Somali, and Eritrean communities, utilizing the specific assets and pathways identified in this study. This work will directly contribute to EHE goals in King County by addressing early HIV testing barriers through culturally responsive, asset-based approaches that can serve as a model for other priority jurisdictions working with immigrant populations.

## Supporting information

**S1 Supplementary Text. Key informant interview guide used in the Harambee 2.0 study.**
(DOCX)

## Author contributions

**Conceptualization:** Rahel Schwartz, Beyene Tewelde Gebreselassie, Farah Mohamed, Mohamed Shidane, Sophia Benalfew, Bethel Tadesse, Hirut Amsalu Libneh, Kifleyesus Bayru, Yikealo K Beyene, Luwam Gabreselassie, Deepa Rao, Roxanne P. Kerani, Rena C. Patel.

**Data curation:** Najma Sheikh, Guiomar Basualdo, Rahel Schwartz, Beyene Tewelde Gebreselassie, Farah Mohamed, Yikealo K Beyene, Luwam Gabreselassie, Deepa Rao, Roxanne P. Kerani, Rena C. Patel.

**Formal analysis:** Najma Sheikh, Guiomar Basualdo, Rahel Schwartz, Farah Mohamed, Roxanne P. Kerani, Rena C. Patel.

**Funding acquisition:** Rahel Schwartz, Farah Mohamed, Sophia Benalfew, Roxanne P. Kerani, Rena C. Patel.

**Investigation:** Guiomar Basualdo, Rahel Schwartz, Farah Mohamed, Mohamed Shidane, Sophia Benalfew, Kifleyesus Bayru, Deepa Rao, Roxanne P. Kerani, Rena C. Patel.

**Methodology:** Ahmed Ali, Rahel Schwartz, Farah Mohamed, Mohamed Shidane, Bethel Tadesse, Hirut Amsalu Libneh, Kifleyesus Bayru, Deepa Rao, Roxanne P. Kerani, Rena C. Patel.

**Project administration:** Ahmed Ali, Rahel Schwartz, Farah Mohamed, Sophia Benalfew, Bethel Tadesse, Deepa Rao, Roxanne P. Kerani, Rena C. Patel.

**Resources:** Shukri Ahmed Hassan, Najma Sheikh, Guiomar Basualdo, Rahel Schwartz, Farah Mohamed, Roxanne P. Kerani, Rena C. Patel.

**Software:** Shukri Ahmed Hassan, Najma Sheikh, Guiomar Basualdo.

**Supervision:** Shukri Ahmed Hassan, Rahel Schwartz, Farah Mohamed, Roxanne P. Kerani, Rena C. Patel.

**Validation:** Shukri Ahmed Hassan, Nahom Daniel, Rahel Schwartz, Beyene Tewelde Gebreselassie, Farah Mohamed, Luwam Gabreselassie, Roxanne P. Kerani, Rena C. Patel.

**Visualization:** Nahom Daniel, Rena C. Patel.

**Writing – original draft:** Shukri Ahmed Hassan, Najma Sheikh, Guiomar Basualdo.

**Writing – review & editing:** Shukri Ahmed Hassan, Najma Sheikh, Guiomar Basualdo, Nahom Daniel, Ahmed Ali, Rahel Schwartz, Beyene Tewelde Gebreselassie, Farah Mohamed, Mohamed Shidane, Sophia Benalfew, Bethel Tadesse, Hirut Amsalu Libneh, Kifleyesus Bayru, Yikealo K Beyene, Luwam Gabreselassie, Deepa Rao, Roxanne P. Kerani, Rena C. Patel.

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
