## [Decision Letter · Decision Letter 0]

27 Nov 2024

PONE-D-24-37628Harambee! 2.0: Community resources and resilience factors to leverage for improving HIV testing behaviors among African immigrant communities in Seattle, WashingtonPLOS ONE

Dear Dr. Hassan,

Thank you for submitting your manuscript to PLOS ONE. After careful consideration, we feel that it has merit but does not fully meet PLOS ONE’s publication criteria as it currently stands. Therefore, we invite you to submit a revised version of the manuscript that addresses the points raised during the review process.

We look forward to receiving your revised manuscript.

Kind regards,

Philipos Petros Gile, MA

Academic Editor

PLOS ONE

Journal Requirements:

2. Thank you for stating the following financial disclosure: [This work was made possible by a National Institute of Allergy and Infectious Diseases (NIAID) Center for AIDS Research (CFAR) supplement award (P30 AI027757).]. Please state what role the funders took in the study. If the funders had no role, please state: "The funders had no role in study design, data collection and analysis, decision to publish, or preparation of the manuscript." If this statement is not correct you must amend it as needed. Please include this amended Role of Funder statement in your cover letter; we will change the online submission form on your behalf.

3. We note that your Data Availability Statement is currently as follows: [All relevant data are within the manuscript and its Supporting Information files.] Please confirm at this time whether or not your submission contains all raw data required to replicate the results of your study. Authors must share the “minimal data set” for their submission. PLOS defines the minimal data set to consist of the data required to replicate all study findings reported in the article, as well as related metadata and methods (https://journals.plos.org/plosone/s/data-availability#loc-minimal-data-set-definition). For example, authors should submit the following data: - The values behind the means, standard deviations and other measures reported; - The values used to build graphs; - The points extracted from images for analysis. Authors do not need to submit their entire data set if only a portion of the data was used in the reported study. If your submission does not contain these data, please either upload them as Supporting Information files or deposit them to a stable, public repository and provide us with the relevant URLs, DOIs, or accession numbers. For a list of recommended repositories, please see https://journals.plos.org/plosone/s/recommended-repositories. If there are ethical or legal restrictions on sharing a de-identified data set, please explain them in detail (e.g., data contain potentially sensitive information, data are owned by a third-party organization, etc.) and who has imposed them (e.g., an ethics committee). Please also provide contact information for a data access committee, ethics committee, or other institutional body to which data requests may be sent. If data are owned by a third party, please indicate how others may request data access.

5. We note you have included a table to which you do not refer in the text of your manuscript. Please ensure that you refer to Table 1 in your text; if accepted, production will need this reference to link the reader to the Table.

Reviewers' comments:

Reviewer's Responses to Questions

**Comments to the Author**

1. Is the manuscript technically sound, and do the data support the conclusions?

Reviewer #1: Yes

Reviewer #2: Yes

2. Has the statistical analysis been performed appropriately and rigorously? 

Reviewer #1: Yes

Reviewer #2: N/A

3. Have the authors made all data underlying the findings in their manuscript fully available?

Reviewer #1: Yes

Reviewer #2: No

4. Is the manuscript presented in an intelligible fashion and written in standard English?

Reviewer #1: Yes

Reviewer #2: Yes

5. Review Comments to the Author

Reviewer #1: Very sound research and well-written and thoughtful paper! Just a few minor comments:

1. In the introduction, it might be good to further explain "deficit-" or "deficiency-centered" work.

2. Perhaps share a little bit more about whether and how findings from East African communities can or should be extrapolated to African immigrant communities more broadly.

3. It would be interesting to know if the authors have any thoughts to share on how to dismantle HIV stigma among religious or faith leaders themselves before tasking them with doing this among their congregations, since it is mentioned that these leaders can often perpetuate this stigma.

4. In the discussion section, there is mention of applying a health equity and social justice lens to health promotion in these communities, and some elaboration on that would be appreciated and beneficial.

5. The results and discussion sections can at times be slightly repetitive and may benefit from tightening the writing just a bit.

6. It may be worth mentioning in the limitations section that digital literacy and comfort was a requirement for participation in the virtual focus groups, which may have possibly excluded some from taking part.

7. The first sentence of the limitations section sates "...explicitly pivots away from their lack of..." I am not sure to what the lack is referring - is there a word missing?

Overall, excellent contribution to the research and important recommendations to advance efforts to address a majoy health disparity!

Reviewer #2: The topic is timely and important as a contribution to knowledge. But the manuscript based on qualitative research and data need to present more data to any reader and be more convincing. For example, data analysis was not explicit. And the results failed to provide direct quotes to show areas of disagreements and agreements between KII and FGDs participants, and amongst KII and FGDs participants. There are diverse participants including PLWH (in KII), religious leaders and community members who were of three East African countries, Ethiopia, Somalia and Eritrean.

At times, it was difficult to discern the authors' and participants view from the data.

Attached is my comments and my notes on the manuscript.

6. PLOS authors have the option to publish the peer review history of their article (what does this mean? ). If published, this will include your full peer review and any attached files.

**Do you want your identity to be public for this peer review?** For information about this choice, including consent withdrawal, please see our Privacy Policy .

Reviewer #1: No

Reviewer #2: **Yes: ** Francisca Omorodion

---

## [Author Response · Author response to Decision Letter 1]

17 Feb 2025

Reviewer 1 Comments and Responses: Word Document

1. Comment #1: Introduction

• The sentence: “Here, our qualitative findings further highlight--, state too early. I suggest move to the beginning of the Result section.

Response: We agree with this suggestion and have moved the sentence to the Results section to improve the flow and readability of the manuscript.

2. Comment #2: Methods – Data Collection

• The decision to use KII and FGDs need to be substantiated and located within other qualitative research methods.

Response: We have revised the Methods section to include a rationale for using key informant interviews (KIIs) and focus group discussions (FGDs) as part of our data collection approach. Specifically, we chose KIIs to gather in-depth insights from key stakeholders prior to conducting FGDs. Conducting KIIs from October 2019 through January 2020 allowed us to refine and adapt the FGD guides based on preliminary findings, ensuring relevance and depth in subsequent group discussions. FGDs were conducted from March through April 2020 and were held virtually due to the COVID-19 pandemic. This phased approach provided complementary data, with KIIs offering individual perspectives and FGDs capturing group dynamics and shared experiences. Details on sampling, interviewers/facilitators, procedures, and guide development are included in the supplementary text.

3. Comment #3: Methods – Sampling

• What are the criteria for participants in KII and FGDs?

Response: Thank you for your important comment regarding the criteria for participants in KIIs and FGDs. This information is already included in the Methods section. To summarize, participants for both KIIs and FGDs were selected using purposeful sampling with input from community partners to determine recruitment methods and potential participants. For the KIIs, we specifically sought healthcare professionals, persons living with HIV (PLWH), and religious and other leaders from East African immigrant communities. Recruitment for PLWH was conducted through provider referrals from case management organizations and the UW-affiliated HIV clinic at the county hospital, with the recruitment period spanning from September 2, 2019, to December 20, 2019. Our initial analysis of KII data indicated the importance of including religious leaders and institutions in any HIV-related stigma reduction intervention. As a result, we prioritized these groups for FGDs while also including other community leaders and members. To minimize the risk of inadvertent disclosure or traumatizing conversations, we did not intentionally recruit PLWH for FGDs. Recruitment for FGDs occurred from January 1 to February 28, 2020, and demographic data collected included age, gender, country of birth, occupation, and religious affiliation. Sampling numbers were guided by theme saturation within participant categories. We hope this clarification addresses your comment. If further adjustments are needed, we are happy to revisit this section.

4. Comment #4: Methods – Sampling

• Could KII participants be part of the FGDs or not? And why or why not?

Response: Thank you for your question. KII participants were not intentionally recruited for the FGDs to avoid potential biases that might arise from having individuals who had already shared in-depth personal views in the KIIs. We aimed to capture a broader range of perspectives in the FGDs, including those of community members who had not participated in the KIIs and, thus, had not had a chance to share their perspectives already. We have now added this clarity in the methods section.

5. Comment #5: Methods – Interviewers and Facilitators

• Did the authors not have their criteria for interviewers and FGD facilitators? Relying solely on community partners may come with biases like favoritism.

Response: Indeed, the reviewer raises an important concern about potential sources of bias in participation recruitment using solely community partners. While community partners often have more direct and proximal access to community members who can best inform the research, relying on them solely can foster favoritism or nepotism. This is a limitation of our work, and we have added language accordingly to our limitations paragraph: “Relying on community partners for recruitment introduces potential bias; however, their engagement with community leaders fostered trust and confidence. Generic strategies like flyers or social media are less effective because these communities rely more on interpersonal connections. Future efforts can incorporate diverse recruitment methods to reduce bias.”

6. Comment #6: Methods – Interview/Discussion Procedures

• Is it community’s or group’s preferred language?

Response: We confirm that interviews and discussions were conducted in the groups’ preferred language and have made this edit in the Methods section.

7. Comment #7: Methods – Interview/Discussion Procedures

• “The KIIs were conducted in several locations---. Interviewer and interviewee preference-“

I think Interviewee preference of location must supersedes that of the interviewer.

Response: We agree and have revised the statement to emphasize that interviewee preference for location was prioritized: “The KIIs were conducted in several locations, based on the interviewee preference.”

8. Comment #8: Methods – Interview/Discussion Procedures

• “Audio from KIIs and FGDs was recorded and then translated and transcribed into English.”

I suggest: We recorded the KIIs and FGDs verbatim and were translated and transcribed into English.

Response: We have revised the sentence as suggested for improved clarity and accuracy.

9. Comment #9: Results – Table 1

• We need to mentation Table 1 as reflecting the background/social characteristics of participants.

Response: We have changed the title of Table 1, to “xxx, to ensure this is explicitly reflected.

10. Comment #10: Results – Combining Data

• The authors seemed to combine the data of KII and FGDs. It would be clear to see the similarities and differences, particularly amongst the divergent individuals and differences.

Response: The reviewer raises an important point. While we understand the suggestion to separate the data from the KIIs and FGDs, we have opted to combine the data as these methods were used to complement each other and provide a holistic view of the perspectives within the community. Nonetheless, when differences exist between themes arising from the KIIs and FGDs, we have now highlighted those in our introduction to the results paragraph, just like we do to highlight any differences between the three communities we worked with.

11. Comment #11: Results – Religious Leaders as Gatekeepers

• How did religious leaders see themselves as gate keepers?

Response: We have revised our terminology to refer to religious leaders as facilitators rather than gatekeepers. The role of religious leaders in community health education was briefly touched on in this research. One participant, a 56-year-old male Somali imam/religious leader, shared, "The elders and religious leaders would be the best people to be trained and given this information in order to create awareness within the community." This perspective underscores their influential role in facilitating health education. However, a more in-depth exploration of this theme is undertaken in the ongoing Harambee 3.0 project, which explicitly works more closely with religious leaders. Data from this project is still being analyzed and will be included in future publications.

12. Comment #12: Results – Qualitative Findings

• I think as a qualitative research paper, we need to see what the KII and FGDs participants said differently and differently. Using direct quotes will show the robustness of the findings and boost our understandings of the data.

Response: We apologize for any lack of clarity. Table 2 includes direct quotes from the transcripts. However, if the reviewer is suggesting that incorporating direct quotes within the text would further enhance the understanding of the data, we have now added key quotes to the manuscript to strengthen our findings.

13. Comment #13: Results – Separating Data

• Separation of KII and FGDs data/findings is paramount.

Response: As stated above, while we understand the suggestion to separate the data from the KIIs and FGDs, we have opted to combine the data for analysis as these methods were used to complement each other and provide a holistic view of the perspectives within the community. Instead, we have chosen to highlight were differences or points of divergence occur in themes between the KIIs and FGDs.

14. Comment #14: Results – Views of PLWH vs. Other Groups

• Do PLWH views the same or differ from the of religious leaders and community members.

Response: The perspectives of PLWH, religious leaders, and community members are discussed in the Results section, highlighting any notable differences in views. However, there were some small differences in themes raised by PLWH vs. religious leaders. We have now added this highlight in differences in our summary intro paragraph to the results.

15. Comment #15: Results – Impact of COVID-19

• Be more explicit on how the COVID pandemic added to the differences between KII and FGDs’ data.

Response: Other than the conduct of the data collection materially, we do not think COVID-19 pandemic influenced the themes of the KIIs or FGDs, the people recruited for them, etc. Nonetheless, it is entirely possible as the conduct of FGDs virtually may have influenced the group dynamics differently. We have now included the following in the limitations section: “The COVID-19 pandemic introduced unique challenges to the study, particularly in transitioning to virtual focus groups. While KIIs were conducted in person, the pandemic required the FGDs to be moved online. While we were pleasantly surprised by how many older persons in the communities adapted to virtual FGDs, it is possible that the shift to virtual FGDs may have impacted the dynamics of group interaction. That the overall themes were consistent across both methods may mitigate this concern.”

16. Comment #16: Discussion – Organization

• Start the sections to reflect the first paragraph. Therefore, start with: HIV education in religious organizations, followed by Utilizing existing community resources, and finally Developing and implementing an intervention.

Response: We have reorganized the Discussion section to reflect this suggested structure for improved coherence and emphasis.

17. Comment #17: Discussion – Linking to Past Studies

• Major findings must be linked to past studies in terms of were they are similar and different. This section further depicts the contribution to knowledge and new contribution to knowledge.

Response: We have revised the Discussion section to link major findings to past studies, highlighting similarities and differences, and emphasizing contributions to new knowledge.

18. Comment #18: Conclusion

• In conclusion, our findings show the significant challenges ----, which can be addressed --- in these studies communities, Ethiopian, Somali and Eritrean.

Response: We have revised the conclusion section opening sentence to reflect your edits above.

19. Comment #19: Conclusion

• How would the pandemic define and shape the findings and health promotion activities in the communities. Would the approach be the same or differ in terms of placed preference.

Response: The reviewer raises astute points about how the COVID-19 pandemic has generally changed conversations among communities of color and immigrant communities, especially in the context of health promotion activities. We can speculate how these same community members may respond to our KII or FGD guides 5 years after the COVID-19 pandemic, however, we do not believe we are in position to include these in our manuscript. Some of our team members have conducted qualitative work in communities of color around the confluence of COVID-19 and HIV for their communities, especially in regards to structural racism and social determinants of health; this work’s first manuscript is still under development.

20. Comment #20: Conclusion – Stigma Reduction

• What about stigma reduction amongst PLWH participants?

Response: The reviewer is raising an important consideration—internalized stigma among PLWH. Indeed, many studies have found this to be an issue among PLWH.1–3 One of the coauthors on our work, Dr. Deep Rao, in fact has developed and implemented stigma reduction interventions among PLWH.1 Our team’s overall emphasis in our work is the reduction of community-level stigma, which is the main barrier in people getting testing for HIV in the first place.4–6 So, while stigma reduction interventions among PLWH, just like among healthcare providers, etc., are important and have their own important roles to play, our project tends to focus on the community side of the larger puzzle, so to speak.

Reviewer 1 Comments and Responses: Email

21. Comment #1: Introduction

• In the introduction, it might be good to further explain "deficit-" or "deficiency-centered" work.

Response: Thank you for this important point, as our position to this may be new for some readers. We have now revised the introduction to provide a clearer explanation of "deficit-" or "deficiency-centered" work to ensure that readers understand the concept. Specifically, we clarify that such an approach typically focuses on identifying and emphasizing the problems or shortcomings within a community, often without considering the strengths, resources, and resilience factors that can be leveraged for change. In contrast, our work critically pivots away from this perspective by focusing on the assets and existing community resources that can support and enhance health interventions, thus empowering the communities rather than viewing them solely through a lens of need or deficiency.

22. Comment #2: Extrapolation to Broader African Immigrant Communities

• Perhaps share a little bit more about whether and how findings from East African communities can or should be extrapolated to African immigrant communities more broadly.

Response: Thank you for this feedback. We have expanded the discussion to explore whether and how the findings from East African communities can be applied to other African immigrant communities, providing context for broader applicability. This study was conducted in exclusive partnership with the Ethiopian, Somali, and Eritrean communities in King County. However, the commonalities in cultural, social, and health-related experiences between these three groups suggest that the findings may be transferable to other African immigrant communities, or even broader immigrant groups, who share similar challenges and resilience factors. We further elaborate on how these shared characteristics can inform health interventions aimed at improving access to HIV testing and care within immigrant populations more broadly. We now state: “Furthermore, while the research was conducted in exclusive partnership with these three East African communities, the shared cultural, social, and health-related experiences among them suggest that the insights gained could have broader applicability. The commonalities observed in the ways these communities approach health, particularly around stigma and access to care, may resonate with other African immigrant populations facing similar barriers. Additionally, immigrant groups from different regions who experience parallel challenges related to acculturation, health disparities, and access to services may benefit from the strategies and interventions developed from this study. This potential transferability underscores the importance of understanding community-specific factors while recognizing broader systemic issues that influence health behaviors and outcomes in immigrant populations. Further research is needed to explore how these findings can inform health interventions for a wider array of immigrant groups.”

23. Comment #3: Dismantling HIV Stigma Among Religious Leaders

• It would be interesting to know if the authors have any thoughts to share on how to dismantle HIV stigma among religious or faith leaders themselves before tasking them with doing this among their congreg

---

## [Decision Letter · Decision Letter 1]

18 Jun 2025

PONE-D-24-37628R1Harambee! 2.0: Community resources and resilience factors to leverage for improving HIV testing behaviors among African immigrant communities in Seattle, WashingtonPLOS ONE

Dear Dr. Hassan,

Thank you for submitting your manuscript to PLOS ONE. After careful consideration, we feel that it has merit but does not fully meet PLOS ONE’s publication criteria as it currently stands. Therefore, we invite you to submit a revised version of the manuscript that addresses the points raised during the review process. Please submit your revised manuscript by Aug 02 2025 11:59PM. If you will need more time than this to complete your revisions, please reply to this message or contact the journal office at plosone@plos.org . Please include the following items when submitting your revised manuscript:

We look forward to receiving your revised manuscript.

Kind regards,

Philipos Petros Gile, MA

Academic Editor

PLOS ONE

Journal Requirements:

Reviewers' comments:

Reviewer's Responses to Questions

**Comments to the Author**

1. If the authors have adequately addressed your comments raised in a previous round of review and you feel that this manuscript is now acceptable for publication, you may indicate that here to bypass the “Comments to the Author” section, enter your conflict of interest statement in the “Confidential to Editor” section, and submit your "Accept" recommendation.

Reviewer #1: All comments have been addressed

Reviewer #3: All comments have been addressed

2. Is the manuscript technically sound, and do the data support the conclusions?

Reviewer #1: Yes

Reviewer #3: Yes

3. Has the statistical analysis been performed appropriately and rigorously? 

Reviewer #1: N/A

Reviewer #3: N/A

4. Have the authors made all data underlying the findings in their manuscript fully available?

Reviewer #1: Yes

Reviewer #3: Yes

5. Is the manuscript presented in an intelligible fashion and written in standard English?

Reviewer #1: Yes

Reviewer #3: Yes

6. Review Comments to the Author

Reviewer #1: Thank you so much for addressing my comments in such a thoughtful manner! Excellent work and I am glad to recommend for publication!

Reviewer #3: (No Response)

7. PLOS authors have the option to publish the peer review history of their article (what does this mean? ). If published, this will include your full peer review and any attached files.

**Do you want your identity to be public for this peer review?** For information about this choice, including consent withdrawal, please see our Privacy Policy .

Reviewer #1: No

Reviewer #3: **Yes: ** Gloria Aidoo-Frimpong

---

## [Author Response · Author response to Decision Letter 2]

31 Jul 2025

1. Introduction Comments:

• The introduction clearly sets out the rationale for the study and situates it within ongoing public health priorities (EHE). The pivot away from deficit-focused narratives is well articulated and timely. The authors provide strong epidemiological grounding specific to King County and African immigrants.

Response: We thank the reviewer for this positive feedback. We are pleased that our introduction effectively situates the study within EHE priorities and that our shift toward strength-based rather than deficit-focused narratives was well-received. We appreciate the recognition of our locally-specific epidemiological grounding for King County and African immigrant communities.

• The introduction could benefit from a slightly stronger framing of how this study builds on existing frameworks for reducing stigma and HIV prevention. A more direct reference to implementation science or community-engaged intervention literature would help position the novelty.

Response: We thank the reviewer for this constructive feedback. We agree that strengthening our theoretical grounding would enhance the introduction's impact. We have revised to more explicitly reference implementation science frameworks (such as the Consolidated Framework for Implementation Research) and community-engaged intervention literature that guides stigma reduction and HIV prevention efforts. We have also better articulated how our asset-based approach builds upon established community-engaged research principles, positioning our work's novel contribution within this broader scholarly context.

• The mention of Harambee! 1.0 is helpful; it might be worth adding a sentence clarifying how Harambee! 2.0 differs from or extends that work.

Response: We thank the reviewer for this suggestion. We agree that clarifying the distinction between our two projects would be helpful for readers. We have add a sentence explaining that while Harambee! 1.0 identified stigma as a barrier through service delivery, Harambee! 2.0 extends this work by conducting in-depth qualitative exploration of stigma mechanisms while simultaneously identifying community assets and resilience factors using an explicitly strength-based framework.

2. Methods Comments:

• The justification for using both KIIs and FGDs is well explained and appropriate. The attention to language, positionality, and consent procedures is commendable and demonstrates ethical rigor. The sampling strategy and community partnership approach are strengths of this work.

Response: We thank the reviewer for this positive feedback on our methodological approach. We are pleased that our rationale for combining KIIs and FGDs was clear and that our attention to language considerations, researcher positionality, and consent procedures was recognized. We appreciate the acknowledgment of our sampling strategy and community partnership approach, as these were central to ensuring culturally responsive and ethically sound research practices.

• Consider clarifying how divergent perspectives across KIIs and FGDs (or between PLWH and other groups) were systematically captured in analysis (beyond noting this in the Results). Were comparison queries or any analytic memos used in NVivo?

Response: We thank the reviewer for this methodological clarification. We agree that our current description does not adequately detail how we systematically captured divergent perspectives across data collection methods and participant groups. We have revised the data analysis section to clarify that we used NVivo's comparison queries to systematically examine differences between KII and FGD responses, as well as between PLWH and other participant categories. Additionally, we maintained analytic memos throughout the coding process to document emerging patterns of convergence and divergence across these groups, which informed our thematic analysis meetings and helped ensure that minority or contradictory perspectives were not overlooked in our final themes.

• The discussion of language (“preferred” vs. “dominant”) is very thoughtful but slightly long. Consider tightening to improve flow.

Response: We thank the reviewer for this feedback. We have condensed this section to maintain the key concepts while improving readability, focusing on the essential distinctions between "preferred language" (what participants are most comfortable using), "dominant language" (English as the language of power), and our intentional avoidance of "limited English proficiency" terminology that frames non-English preference as deficient.

3. Results Comments:

• The results are well structured and thematically clear. The use of direct participant quotes is strong and gives voice to the communities. Table 2 is an excellent addition and provides good transparency.

Response: We thank the reviewer for this positive feedback on our results presentation. We are pleased that our thematic structure was clear and that the direct participant quotes effectively amplified community voices, this was a priority for us in ensuring authentic representation of participants' experiences. We also appreciate the recognition of Table 2's contribution to transparency, as we felt it was important to provide readers with clear insight into our analytical process and thematic development.

• While the decision to combine KII and FGD data is justified, further explicit signposting of where key divergences occurred (especially between PLWH vs. religious leaders, or FGDs vs. KIIs) would enrich the depth of analysis.

Response: We thank the reviewer for this important observation. We have addressed this feedback by explicitly stating any key divergences throughout our revised Results section.

• Some subthemes could be better separated. For example, the section on leveraging PLWH as educators could be elevated more clearly as a distinct theme or subtheme.

Response: We agree with this suggestion and have made necessary edits. The concept of leveraging PLWH as community educators represents a distinct and important finding that deserves greater prominence.

• Consider integrating a brief summary paragraph at the end of the Results, highlighting key points of agreement vs. divergence across communities and participant types.

Response: We thank the reviewer for this excellent suggestion. We have added a summary paragraph at the end of the Results section that synthesizes the key points of convergence and divergence across the three communities and between participant types. This will provide readers with a clear analytical synthesis before moving to the Discussion.

4. Discussion Comments:

• The Discussion is significantly strengthened in this revision, well-structured, and now more clearly linked to prior literature. The section on applying a health equity and social justice lens is particularly well done. The implications for practice are thoughtful and actionable.

Response: We thank the reviewer for this positive feedback on our revised Discussion section. We are pleased that the health equity and social justice framing was well-received and that our practice implications were found to be actionable.

• There is still some residual repetition between the Results and Discussion. the first paragraphs under each subheading in the Discussion could be tightened.

Response: We agree that tightening the opening paragraphs of each Discussion subheading would improve flow and reduce redundancy. We have revised these sections to focus on interpretation and implications rather than restating findings, ensuring the Discussion adds analytical value beyond what was presented in the Results.

• The contribution to broader HIV stigma reduction and implementation science literature could be more explicitly articulated. For example, how does this advance multilevel stigma reduction frameworks? How might this inform the adaptation of interventions for other immigrant groups?

Response: We thank the reviewer for this important suggestion. We have added a section to the discussion of how our findings advance multilevel stigma reduction frameworks, particularly the Earnshaw model we referenced, and how our asset-based approach could inform adaptation of interventions for other immigrant populations. We have also better articulated our contributions to implementation science literature regarding community-engaged approaches in stigmatized health conditions.

• The limitations section is improved. Consider briefly mentioning potential social desirability bias in community-based FGDs on a sensitive topic like HIV.

Response: We thank the reviewer for this suggestion and have added a sentence to highlight this bias.

5. Conclusion Comments:

• The Conclusion is clear and reinforces the study’s key messages.

Response: We thank the reviewer for this positive feedback. We are pleased that our conclusion effectively reinforces the study's key messages and provides clear synthesis of our findings.

• The focus on community assets and resilience is well aligned with the framing of the paper.

Response: We appreciate this recognition of our consistent framing throughout the paper.

• Consider adding a forward-looking sentence about potential next steps (e.g., developing and piloting an intervention based on these findings, future partnerships, contribution to EHE goals).

Response: We thank the reviewer for this excellent suggestion. We have added a forward-looking sentence to our conclusion that outlines our next steps, including plans to develop and pilot an intervention based on these findings in partnership with the same communities, and how this work will contribute to EHE goals in King County.

---

## [Editor Report · Decision Letter 2]

24 Aug 2025

Harambee! 2.0: Community resources and resilience factors to leverage for improving HIV testing behaviors among African immigrant communities in Seattle, Washington

PONE-D-24-37628R2

Dear Author,

We’re pleased to inform you that your manuscript has been judged scientifically suitable for publication and will be formally accepted for publication once it meets all outstanding technical requirements.

Kind regards,

Philipos Petros Gile, MA

Academic Editor

PLOS ONE
---

## [Editor Report · Acceptance letter]

PONE-D-24-37628R2

PLOS ONE

Dear Dr. Hassan,

I'm pleased to inform you that your manuscript has been deemed suitable for publication in PLOS ONE. Congratulations! Your manuscript is now being handed over to our production team.

Kind regards,

on behalf of

Dr. Philipos Petros Gile

Academic Editor

PLOS ONE